

# L1 and L2 reading skills in Dutch adolescents with a familial risk of dyslexia

Ellie R.H. van Setten[1,2], Wim Tops[1], Britt E. Hakvoort[3], Aryan van der Leij[3], Natasha M. Maurits[2,4] and Ben A.M. Maassen[1,2]

[1] Center for Language and Cognition Groningen (CLCG), Faculty of Arts, University of Groningen, Groningen, the Netherlands
[2] Research School of Behavioural and Cognitive Neurosciences (BCN), University Medical Center Groningen, University of Groningen, Groningen, the Netherlands
[3] Research Institute of Child Development and Education, University of Amsterdam, Amsterdam, the Netherlands
[4] Department of Neurology, University Medical Center Groningen, University of Groningen, Groningen, the Netherlands

Corresponding author
Ellie R.H. van Setten,
e.r.h.van.setten@rug.nl

## ABSTRACT

**Background**. The present study investigated differences in reading and spelling outcomes in Dutch and English as a second language (ESL) in adolescents with a high familial risk of dyslexia, of whom some have developed dyslexia (HRDys) while others have not (HRnonDys), in comparison to a low familial risk control group without dyslexia (LRnonDys). This allowed us to investigate the persistence of dyslexia in the first language (L1) and the effect of dyslexia on the second language (L2), which has, in this case, a lower orthographic transparency. Furthermore, the inclusion of the HRnonDys group allowed us to investigate the continuity of the familial risk of dyslexia, as previous studies observed that the HRnonDys group often scores in between the HRDys and LRnonDys group, and whether these readers without reading deficits in Dutch, have more reading difficulties in ESL.

**Methods**. The data of three groups of adolescents were analyzed; 27 LRnonDys, 25 HRdys 25 HRnonDys. The mean age was 14;1 years; months, and 37 were male. All were native speakers of Dutch, attended regular secondary education (grade 7–10), and were non-native speakers of English. Using MANOVA the groups were compared on Dutch and English word reading fluency (WRF), spelling and vocabulary, Dutch pseudoword and loanword reading fluency, phonological awareness (PA), rapid automatized naming (RAN), and verbal short term and working memory. A repeated measures ANOVA was used to compare English and Dutch WRF, spelling and vocabulary directly within the three groups.

**Results**. The analyses revealed that the HRDys group had a deficit in both reading and spelling in Dutch and ESL. They also performed poorer than the LRnonDys group on all other measures. Effect sizes were especially large for pseudoword reading and the reaction times during the PA task. The HRnonDys group scored generally poorer than the LRnonDys group but this difference was only significant for Dutch pseudoword reading, PA reaction times and verbal short term memory. In general the HRDys and HRnonDys group scored similar in Dutch and English, except for English WRF where the HRDys group scored slightly better than expected based on their Dutch WRF.

**Discussion**. There was a high persistence of dyslexia. Adolescents with dyslexia had large impairments in reading and spelling, and reading related measures, both in Dutch and ESL. Despite high inter-individual differences, an overall three-step pattern was observed. Adolescents in the HRnonDys group scored in between the HRDys and LRnonDys group, supporting the polygenetic origin of dyslexia and the continuity of the familial risk of dyslexia. The lower orthographic transparency did not have a negative effect on L2 reading, spelling and vocabulary, both in the HRnonDys and HRDys group. The latter group performed slightly better than expected in L2, which may be a result of the massive exposure to English and high motivation to use English by adolescents.

## INTRODUCTION

Dyslexia is a specific learning disorder characterized by problems with accurate and/or fluent word recognition, poor decoding, and poor spelling abilities at the word level (*American Psychiatric Association, 2013*, p. 67), but it can have consequences for higher level reading comprehension skills and writing skills as well (*Tops et al., 2013*). Dyslexia is not a discrete disorder, there is no natural cut-off point that separates the reading skills of people with and without dyslexia (*Shaywitz et al., 1992*). Furthermore, dyslexia is a heterogeneous disorder, as research has found that there are multiple genes and cognitive deficits that may affect the reading skills of people with dyslexia (*Carrion-Castillo et al., 2017*; *Carrion-Castillo, Franke & Fisher, 2013*; *Pennington, 2006*; *Pennington et al., 2012*). While a phonological processing deficit, especially a deficit in phonological awareness (PA), is often seen as the core problem of dyslexia (e.g., *Ramus et al., 2003*; *Van der Leij & Morfidi, 2006*), it is neither necessary nor sufficient to explain all cases of dyslexia at the individual level (*Pennington et al., 2012*). Other cognitive deficits linked to reading problems include deficits in rapid automatized naming (RAN; e.g., *Kirby et al., 2010*; *Norton & Wolf, 2012*; *Papadopoulos, Spanoudis & Georgiou, 2016*), verbal working and short-term memory (VST/WM; e.g., *De Jong, 1998*; *Pennington & Lefly, 2001*; *Snowling, Gallagher & Frith, 2003*) and orthographic processing (e.g., *Georgiou et al., 2012*; *Rothe et al., 2015*).

Dyslexia has a prevalence of three to ten percent in the normal population, depending on the criteria used (*Miles, 2004*). However, as dyslexia has a partially genetic origin (e.g., *Carrion-Castillo, Franke & Fisher, 2013*) children with a parent with dyslexia have a much higher risk to develop dyslexia, on average 45 percent (*Snowling & Melby-Lervåg, 2016*). Because we can reliably predict, before formal reading instruction starts, that a large proportion of children with a (high) familial risk of dyslexia (FR) will develop dyslexia, they constitute an interesting population for prospective longitudinal research. The participants of the present study have all participated in such a study in the Netherlands, the Dutch Dyslexia Program (DDP; See e.g., *Van Bergen et al., 2012*; *Van der Leij et al., 2013*). In the DDP study, a group of children, of whom approximately two-third has a high FR and

half of the children with a high FR has eventually developed dyslexia, has been followed from birth. The present study focused on a selection of these children at the age when they were advanced readers. More specifically, we investigated how (a family risk of) dyslexia influences the word level reading and spelling skills in early adolescence. As English is a mandatory subject in schools in the Netherlands, and is increasingly used in everyday life, we did not only examine reading and spelling in the participants' first language (L1) Dutch, but also reading and spelling in English, their second language (L2). To our knowledge this is the first large-scale study that involves adolescents with a high FR that has looked at L2 reading and spelling. Before we describe our research questions and hypotheses in more detail, we will first discuss general differences that have been found between (advanced) readers with and without (a familial risk of) dyslexia, the position of English in the Netherlands, and learning to read in English as a second language (ESL) with dyslexia.

## Readers with a familial risk of dyslexia

As dyslexia is not a discrete disorder, the degree at which parents are affected by dyslexia also varies. Hence, the familial risk for dyslexia is continuous as well. Some of the cognitive and reading impairments typically associated with dyslexia have also been found, usually in a milder form, in children with a high FR who do not have dyslexia (HRnonDys). In family risk studies they often score in between the group with a high FR with dyslexia (HRDys) and children with a low FR without dyslexia (LRnonDys) in family risk studies (e.g., *Elbro, Borstrøm & Petersen, 1998*; *Pennington & Lefly, 2001*; *Snowling, Gallagher & Frith, 2003*; *Van Bergen et al., 2012*). This three-step pattern is in line with a polygenetic inheritance and multifactorial origin of dyslexia, as the HRnonDys group probably has inherited or has been exposed to some risk factors associated with reading disability causing mild reading problems (see *Van Bergen, Van der Leij & De Jong, 2014* for a more in-depth discussion).

For advanced readers, results regarding this three-step pattern are somewhat mixed. *Snowling, Muter & Carroll (2007)* found significant differences in exception word reading, text reading accuracy, and timed measures of nonsense passages, sight-word and decoding efficiency, text reading rate, and spelling, between HRnonDys adolescents and a control group of adolescents with a low familial risk without dyslexia (LRnonDys) at the age of 12–13. On the other hand, *Dandache, Wouters & Ghesquière (2014)* found only a difference for Word Reading Fluency (WRF) in Dutch readers in grade 6, and *Eklund et al. (2014)* did not find significant difference between the HRnonDys group and the LRnonDys group in Finnish readers in grade 8. However, in all studies the scores of the HRnonDys group were generally lower than the scores of the LRnonDys group, and higher than the scores of the HRDys group. In all studies, the HRDys group also scored significantly lower than the LRnonDys control group. These effects were usually large, supporting the view that dyslexia persists into adolescence, as has also been found by others (*Ferrer et al., 2015*; *Shaywitz et al., 1999*).

## Position of English in the Netherlands

Although Dutch is the dominant language in the Netherlands, English has a very important position (See for a more elaborate discussion of the position of English in the Netherlands

also: *Gerritsen et al., 2016*). According to *McArthur (1996)* the high level of bilingualism between English and Dutch in the Netherlands justifies the view that English is a strong second language, rather than a foreign language. In the Netherlands, children learn to read and write in their first language Dutch when they are six years old and enter grade 1. Explicit English L2 instruction is mandatory from grade 5 onwards and remains obligatory in almost all curricula until the end of secondary education and even in a wide range of fields of study in higher education. However, some schools already start English as early as Kindergarten, and the hours of English per week vary (*Thijs et al., 2011*). Dutch children come into contact with English almost daily via television and multimedia, like the internet and games. According to *Di Carlo (1994)* contact with multimedia, where verbal messages are accompanied by visual images, has an impact on language learning. Moreover, it has been shown that watching subtitled programs and films in a foreign language causes (incidental) language learning, even in young children (*D'Ydewalle & Poel, 1999*) because both languages are then simultaneously present.

## Reading in English (as a second language) with dyslexia

While English and Dutch both stem from Germanic origins, and share a relatively high syllabic complexity, they differ in the transparency of their letter-to-sound conversions. Dutch orthography has 40 phonemes that can be written in 76 ways, in contrast to English that has 44 phonemes that can be written in 561 ways (*Dewey, 1971*). Dutch orthography is therefore considered more transparent than English. In transparent orthographies, firm and accurate phonological representations are installed in the early stages of reading acquisition. Therefore, phonological awareness and reading development are faster in transparent than in deeper orthographies (*Seymour, Aro & Erskine, 2003*). This explains why a large proportion of Dutch children can read accurately after one year of reading instruction, whereas English children need on average four years to obtain the same reading level (*Seymour, Aro & Erskine, 2003*).

According to the Linguistic Coding Differences Hypothesis (LCDH; *Ganschow et al., 1991*) individuals who encounter difficulties with L1 acquisition, will also have problems in L2 because there is a link in L1 and L2 learning, particularly in learning to read. *Ganschow et al. (1991)* emphasized the importance of orthographic, syntactic and semantic, and especially phonological skills to transition from L1 to L2. Although the relation between phonological awareness and reading decreases with age, phonological processing still plays a role in the reading skills of individuals with dyslexia at a later age (*Bekebrede, Van der Leij & Share, 2009*). For most typically developing children, the transition from L1 to L2 learning runs smoothly, whereas for children with dyslexia, L2 learning is particularly challenging (*Helland & Kaasa, 2005*). In transparent orthographies, children with dyslexia encounter difficulties with reading speed. In deeper orthographies, such as English, children with dyslexia experience difficulties with reading accuracy as well (*Hagtvet & Lyster, 2003*).

In addition to phonological processing, orthographic processing is also necessary for fluent reading. There is much evidence for differences in orthographic knowledge between typical and dyslexic readers (*Bekebrede, Van der Leij & Share, 2009*). However, there is also evidence that a subgroup of readers with dyslexia read better in English L2 than in their

more transparent L1. *Miller-Guron & Lundberg (2000)* found that a group of dyslexic adult students that had a self-reported preference for English (L2) over Swedish (L1), indeed read better in English than in Swedish. Furthermore, the dyslexic students with a preference for English also performed better on English spelling and free-writing tasks, compared to dyslexic students without a preference for English. *Miller-Guron & Lundberg (2000)* have suggested that the dyslexic participants with a preference for English could perhaps make use of frequently occurring spelling patterns or orthographic structures that they learned from frequent exposure to English. According to the authors, irregular grapheme-to-phoneme conversion rules in English dissuade readers from using the decoding-route and favor the use of whole-word strategies. Additionally, *Van der Leij & Morfidi (2006)* found that a subgroup of Dutch (L1) poor readers in grade 7 and 8 were better at English (L2) reading than at Dutch reading, and had a higher verbal competence in English than they had in their mother tongue. This group was shown to have superior orthographic competence, as measured with an orthographic choice task in both Dutch and English, compared to a group of dyslexic readers who failed to show an advantage in L2 English.

## The present study

The present study focused on Dutch and ESL literacy skills in Dutch adolescents with a familial risk of dyslexia. As dyslexia is a problem at the word level we compared the three main groups, HRDys, HRnonDys and LRnonDys, on word level reading and spelling, in Dutch and English, as well as on word-level reading related skills: RAN, PA, vocabulary and VST/WM. Based on previous studies we expected large differences between the HRDys group and the LRnonDys group on all tests, as dyslexia is a persistent problem. In line with previous studies we expected that the HRnonDys group will perform in between the HRDys and LRnonDys group, but that these differences may not always be significant. As L1 reading skills are highly predictive for L2 reading skills according to the LCDH we expected that reading and spelling in ESL is also highly problematic for the HRDys group. To investigate if the effects are similar across languages, we directly compared Dutch and English reading, spelling and vocabulary skills between the groups. The lower orthographic transparency may make reading in English more difficult, but a preference for reading in ESL has also been observed among some dyslexic readers. Moreover, the inclusion of a HRnonDys group allowed us to investigate if this group, which by definition has no reading problems in Dutch, does have reading difficulties in English which is orthographically more complex than Dutch.

## METHOD

### Participants

Eighty-one adolescents were included in the present study, 39 were male and 42 were female. They attended year 1 to 4 of Dutch secondary school, corresponding to grades 7–10 (mostly grade 8), and the mean age was 14;1 years; months ($SD = 7$ months). They all participated in the Dutch Dyslexia Program (DDP) in Groningen or Amsterdam, the Netherlands. Seventy-eight percent of the children from Amsterdam and Groningen who participated in the previous DDP measurement in grade 6, participated in the present

**Table 1  Participant characteristics.**

| Group | n | n males | Age (months) | | School grade | |
|---|---|---|---|---|---|---|
| | | | M | SD | M | SD |
| LRnonDys | 27 | 11 | 166.9 | 5.3 | 8.1 | .5 |
| LRDys | 4 | 2 | 163.5 | 4.4 | 7.8 | .5 |
| HRnonDys | 25 | 12 | 170.0 | 9.0 | 8.1 | .7 |
| HRDys | 25 | 14 | 170.7 | 6.8 | 8.1 | .9 |

Notes.

LRnonDys, Low Risk without Dyslexia; LRDys, Low Risk with Dyslexia; HRnonDys, High Risk without Dyslexia; HRDys, High Risk with Dyslexia.

study. All participants were native speakers of Dutch and spoke mainly Dutch in school, but a few participants in the Northern region also spoke Frisian at home. None of the participants had lived abroad in an English speaking country other than during a holiday.

Based on parental reading scores participants were divided into a high FR and low FR control group. Parents were tested with WRF and pseudoword reading fluency (PWRF) tasks, the same tasks that were used in this study and are described later. If the reading scores of at least one parent belonged to the lowest 20 percent on one test and to the lowest 40 percent on the other test, using the norms by *Kuijpers et al. (2003)*, and if they reported a family history of dyslexia, participants were categorized in the high FR group. For the low FR control participants, reading scores of both parents did not meet these criteria, nor was there a self-reported family history of dyslexia. Dyslexia status was based on the participants' reading scores during previous DDP measurements. The adolescents were categorized in the group with dyslexia if they met the following criteria: participation in at least two out of three measurements in grades 2, 3, and 6, two scores in the lowest 10% range on either WRF or PWRF (again measured with the same tests), and at least below average on the other test during that measurement. If they had participated only twice and met the criteria once they were excluded because we aimed to include only participants with persistent reading problems (one participant was excluded). By combining the risk and dyslexia categorizations, four groups were created: 27 participants with a low FR without dyslexia (LRnonDys), 25 participants with a high FR without dyslexia (HRnonDys), 25 participants with a high FR with dyslexia (HRDys) and four participants with a low FR with dyslexia (LRDys). While the latter group is too small for group comparisons, these adolescents were included in the standardization of the test measures, which was based on the whole low FR group, to have a more accurate estimation of the typical population. Participant characteristics can be found in Table 1. Because of the longitudinal design groups could not be matched on factors like age and gender. As age did not differ significantly between the three main groups ($F$ (2, 74) = 2.085, $p$ = .132), we did not further consider age in the analyses.

Exclusion criteria were an IQ below 80 as measured during previous assessments, severe medical or psychiatric problems, or the attendance of a secondary school where the language of instruction for other subjects than English was English. For the latter reason, four participants (two in each non-dyslexic group) were excluded from the study. Since

the other exclusion criteria were already applied during previous DDP assessments no participants were excluded for these reasons. Participants with comorbid developmental disorders, such as Attention Deficit Hyperactivity Disorder (ADHD), Attention Deficit Disorder (ADD) or Autism Spectrum Disorder (ASD) were included in the study as comorbidity is common between dyslexia and other (neuro)-developmental problems (e.g., *Germanò, Gagliano & Curatolo, 2010*; *Kaplan et al., 1998*; *Mayes & Calhoun, 2006*; *Willcutt & Pennington, 2000*), and excluding these participants would lead to an unrepresentative sample. Parents reported three cases of comorbid disorders in the LRnonDys group (one with ASD and two with ADD), two in the HRnonDys group (one with ASD and one with ADD), and four in the HRDys group (one with ASD, two with ADHD, and one with ADD). Parents reported that one participant in the LRDys group showed signs of both ADD and ASD, but it was unclear whether this participant had an official diagnosis. From questionnaires conducted in grade 6 we know that about 11 percent of the children in the LRnonDys group, 14 percent of the children in the HRnonDys group, and 75 percent of the children in the HRDys group received extra help in school.

## Materials

### Reading fluency

Four tests of reading fluency, measuring both reading accuracy and speed, were included in the study: Dutch WRF ("Een minuut Test", one minute test, *Brus & Voeten, 1973*), Dutch PWRF ("Klepel", *Van den Bos et al., 1994*), English WRF (*Kleijnen, Steenbeek-Planting & Verhoeven, 2008*), and loan word reading fluency, using loanwords from English that are incorporated in the Dutch language (*Schijf, 2009*). The participants had to read aloud word lists, with words of increasing difficulty, as fast and accurately as possible. The score is the number of words read correctly within the allotted time, which was one minute for all tests, except for PWRF where it was two minutes. The English WRF test had 108 words and the Loan word test 116 words; for each test the words were presented in four columns. For the Dutch WRF and PWRF the A-version was used, but the last column of the B-version was added, such that there were 145 words spread over five columns, to ensure that participants would not finish reading all words within the allotted time.

### Spelling

Dutch and English spelling was measured with a dictation test. First a whole sentence was read, next the target word(s) were repeated. The participant had to write the target words on an answer sheet where the rest of the sentence was already printed. The test was executed at the participant's own pace and at request words were repeated more than once. For Dutch the "PI-signalerings dictee" (*Geelhoed & Reitsma, 2004*) was used, which is a spelling screening instrument for the first grade of secondary school. It consists of 38 sentences and has 90 target words. For Dutch learners of English there was no standardized spelling test available. Therefore we created an English dictation test based on English textbooks for the lowest level of the first two grades of Dutch secondary school (grades 7 and 8). This dictation consists of 20 sentences with 20 target words, and can be found in Appendix A.

### Vocabulary

Dutch and English vocabulary knowledge were tested with an oral vocabulary test. For Dutch this was the vocabulary task from the third edition of the Dutch Wechsler Intelligence Scale for Children (WISC-III-NL, *Wechsler, 2005*). During this task participants were asked to describe the meaning of words that increased in difficulty as the test progressed, like for example "paraplu" (umbrella) and "onafhankelijk" (independent). There are 35 items and two points can be gained for a fully correct answer and one point for a partially correct answer as determined by the test manual. After four incorrect answers the test is terminated. English vocabulary knowledge was tested with the fourth edition of the Peabody Picture Vocabulary Test (PPVT-IV, *Dunn & Dunn, 2007*). During this test the participants heard a word and had to indicate which picture out of four pictures matches with the word. First a base set was established, a set of 12 items where all answers were correct or only one mistake was made. Thus, if more mistakes were made in the starting set, determined by the participant's age, a lower set was chosen until the base set was reached. Next, the ceiling set was established, the set for which eight or more mistakes were made, by moving up from the starting set. A total score is obtained by subtracting the total number of errors from the item number of the last item of the ceiling set. The maximum score for this test is 228.

### Phonological awareness

PA was measured with a Dutch spoonerism task (*Depessemier & Andries, 2009*). The test was computerized and the items were recorded such that all participants heard the items exactly the same way. The participants heard two words through headphones and were instructed to switch the first sounds of the words such that the first sound of the second became the first sound of the first word, and vice versa. This resulted mostly in pseudowords, but sometimes also in other real words. For example "nieuwe wagen" (new wagon) became "wieuwe nagen". If a word started with a consonant cluster, both the answer where the whole cluster was reversed and the answers where a single phoneme was reversed were considered correct. For example in the case of "grote boom" (big tree), both "brote goom" and "bote groom" were considered correct. There were six practice items and 20 test items. For the test items the participants were instructed to respond as fast and accurately as possible, but only if they knew the complete answer, such that the researcher could record both reaction times and accuracy. Reaction times were recorded manually when the participant started to utter the first word. The reaction times scores were corrected for extreme outliers and wrong registrations by removing reaction times shorter than 100 ms and longer than 3 standard deviations above the mean reaction time per participant.

### Rapid automatized naming

RAN was measured with a digit naming task (*Van den Bos & Lutje Spelberg, 2007*). The participant had to name 50 digits grouped in five rows as fast and accurately as possible. Taking into account accuracy and the time needed to name all items the number of items that could be named correctly within one minute was calculated.

### Verbal short term and working memory

The Digit span task of the WISC-II-NL (*Wechsler, 2005*) was used to measure VST/WM. During the forward task, where mainly short term memory is involved, the participants heard a series of digits that had to be repeated in the same order. During the backward task the participant had to repeat the series of digits in a reversed order. Here working memory is also involved as the information does not only have to be stored but also has to be manipulated. The length of the series increases during both digit span tasks. If the participant failed to repeat two series of the same length, the test is terminated. The score is the number of series reported correctly. The maximum score is 16 in the forward task and 14 in the backward task.

### Procedure

Participants were seated in a quiet test environment. Tests were conducted individually. As they participated in the DDP before, participants were already familiar with this testing situation. First, all Dutch tests were conducted, then the reading related subtests, and finally the English tests, to not confuse the participants with the different languages. There was a break in between the reading related tests and the English tests and participants were encouraged to take more breaks if deemed necessary. Informed consent was obtained from the parents, and adolescents gave their informed assent, as well. Travel costs were reimbursed and participants received a gift voucher to thank them for their participation. The study was approved by the Medical Ethical Committee of the University Medical Center Groningen (METc-nr: 2014/076, ABR-nr: NL48140.042.14).

### Analyses

To investigate whether data were missing completely at random, and whether we could use expectation maximization to impute missing data, a missing value analysis was conducted with Little's MCAR test in SPSS 23 (IBM Corp, Armonk, NY, USA). A Multivariate Analysis of Variance (MANOVA) was conducted between the LRnonDys, HRnonDys and HRDys groups to investigate first if there was a group difference over all measures together, and next to investigate for which measures the groups differed from each other. Post-hoc pairwise comparisons with the Least Square Difference Method were used to investigate which groups differed from each other. Since we did overall tests first, and since we have separate hypotheses for each contrast, namely that we would see a three-step pattern for each of the included reading (related) test where the LRnonDys group would have the highest score and the HRDys group the lowest, we did not further correct for multiple comparisons as this would strongly reduce power and increase the Type-II error rate. Hedges' $g$ was calculated as a measure of the effect sizes based on the mean and standard deviations. This adaptation of the commonly reported Cohen's $d$ corrects for a bias due to small sample sizes (*Hedges & Olkin, 1985*). For Dutch and English WRF, vocabulary and spelling, repeated measures ANOVAs with a 3x2 design were used to investigate if the effect of Group (LRnonDys, HRDys, HRnonDys) was larger in a certain Language (Dutch or English). All hypotheses were tested two-sided, but we also report marginally significant effects ($p < .1$) that would be significant if we tested one-sided. If variables were not

**Table 2  Descriptive statistics of standardized scores per group.**

| | LRnonDys ($n=27$) | | | HRnonDys ($n=25$) | | | HRDys ($n=25$) | | |
|---|---|---|---|---|---|---|---|---|---|
| | *M (SD)* | Skewness | Kurtosis | *M (SD)* | Skewness | Kurtosis | *M (SD)* | Skewness | Kurtosis |
| Dutch Words RF | .21 (.78) | −.08 | −1.10 | −.01 (.96) | .11 | .14 | −2.00 (1.14) | −.76 | .25 |
| Pseudoword RF | .24 (.80) | .06 | −.62 | −.38 (.83) | −.37 | .66 | −1.75 (.67) | −.95 | .71 |
| Loanword RF | .21 (.85) | −.31 | .35 | −.21 (.84) | .37 | 1.34 | −1.77 (1.52) | −.42 | .18 |
| English Word RF | .18 (.87) | .08 | −.61 | −.27 (.83) | .66 | 1.53 | −1.56 (1.25) | .03 | .06 |
| Dutch Spelling | .26 (.74) | −.62 | −.13 | −.18 (.99) | −.58 | −.35 | −2.33 (1.94)[a] | −.39 | −1.18 |
| English Spelling | .24 (.79) | −1.19 | 1.99 | −.17 (.77)[a] | −1.48 | 2.71 | −1.93 (.97)[a] | .29 | −.94 |
| Dutch Vocabulary | .07 (1.03) | −2.22 | 7.33 | −.15 (.62) | .36 | −.23 | −.57 (.73) | −.66 | −.44 |
| English Vocabulary | .12 (.98) | −.41 | −.39 | −.07 (.85) | −.69 | .24 | −.66 (1.17) | −.10 | −.69 |
| Spoonerisms - Acc | .16 (.59)[a] | −.71 | −.67 | −.55 (1.40)[a] | −1.19 | .41 | −2.79 (2.39) | −.59 | .99 |
| Spoonerisms - RT (Log) | −.27 (.67) | .73 | .57 | .66 (.88) | −.56 | −.62 | 1.98 (.91) | .73 | .74 |
| RAN (Items/min) | .18 (.94) | .43 | −.56 | −.01 (.95)[a] | .75 | 2.19 | −.58 (.60) | −.24 | −.42 |
| Digit Span - Forward | .11 (1.00)[a] | .15 | −1.14 | −.56 (.74)[a] | −.44 | −.27 | −.95 (.69) | .20 | −.78 |
| Digit Span - Backward | .15 (.98) | −.01 | −.92 | −.21 (1.09) | .26 | .18 | −.75 (.75)[a] | −.20 | −1.10 |

Notes.

[a] the distribution of this variable within this group differed significantly from a normal distribution according to the Kolmogorov–Smirnov test for normality.

LRnonDys, Low Risk without Dyslexia; HRnonDys, High Risk without Dyslexia; HRDys, High Risk with Dyslexia; RF, Reading Fluency; Acc, Accuracy; RT, Reaction Time; RAN, Rapid Automatized Naming.

normally distributed, and/or if the homogeneity of variance assumption was violated we also ran non-parametric analyses, but since the conclusions based on the non-parametric analyses were very similar we only report the results of the parametric analyses.

## RESULTS

### Missing data analysis and descriptive statistics

One case was missing for Dutch spelling, two cases were missing for RAN, and 4 cases were missing for the spoonerism reaction time due to test administration errors, registration or computer errors. Since the data were missing completely at random ($\chi^2(35) = 26.510$, $p = .848$), and the number of missing cases was small, missing cases were imputed with the expectation maximization method in SPSS. Normality within the groups was checked with the Kolmogorov–Smirnov test. In Table 2 with descriptive statistics it is indicated when the distribution deviated significantly from a normal distribution. The data of the Spoonerisms RT, which had a severe positive skewness and kurtosis, was successfully transformed with a log transformation; for the other variables that were not normally distributed transformations were not successful. There was homogeneity of variance for most variables, except for Digit-span - Forward, Spoonerisms Accuracy and RT, and Dutch spelling. To make the scales of the different reading and reading related measures comparable, the data were standardized on the basis of the means and standard deviation of the whole low familial risk group, including 4 participants with dyslexia. Therefore the means and standard deviations of the LRnonDys group slightly deviate from zero and one, respectively. Correlations between the different reading (related) measures can be found in

**Table 3  Pearson correlation between Dutch and English reading and reading related skills for participants in the LRnonDys, HRnonDys and HRDys group.**

| | 1 | 2 | 3 | 4 | 5 | 6 | 7 | 8 | 9 | 10 | 11 | 12 | 13 |
|---|---|---|---|---|---|---|---|---|---|---|---|---|---|
| 1. Dutch Words RF | – | | | | | | | | | | | | |
| 2. Pseudoword RF | .88 | – | | | | | | | | | | | |
| 3. Loanword RF | .87 | .81 | – | | | | | | | | | | |
| 4. English Word RF | .81 | .75 | .89 | – | | | | | | | | | |
| 5. Dutch Spelling | .76 | .71 | .77 | .70 | – | | | | | | | | |
| 6. English Spelling | .80 | .79 | .80 | .77 | .84 | – | | | | | | | |
| 7. Dutch Vocabulary | .44 | .37 | .48 | .52 | .45 | .55 | – | | | | | | |
| 8. English Vocabulary | .43 | .39 | .59 | .71 | .43 | .51 | .36 | – | | | | | |
| 9. Spoonerisms - Accuracy | .62 | .60 | .68 | .66 | .65 | .69 | .30 | .58 | – | | | | |
| 10. Spoonerisms - RT (Log) | −.68 | −.74 | −.63 | −.59 | −.65 | −.63 | −.32 | −.30 | −.61 | – | | | |
| 11. RAN (Items/min) | .58 | .60 | .48 | .38 | .34 | .38 | .17 | .19 | .25 | −.39 | – | | |
| 12. Digit Span - Forward | .44 | .43 | .35 | .36 | .41 | .44 | .24 | .26 | .40 | −.41 | .25 | – | |
| 13. Digit Span - Backward | .46 | .40 | .34 | .29 | .50 | .42 | .25 | .15 | .28 | −.32 | .36 | .59 | – |

**Notes.**

$N = 77$.

LRnonDys, Low Risk without Dyslexia,; HRnonDys, High Risk without Dyslexia,; HRDys, High Risk with Dyslexia,; RF, Reading Fluency,; RT, Reaction Time,; RAN, Rapid Automatized Naming. Correlations of .19 and larger are significant at .1 level (2-tailed), Correlations of .24 and larger are significant at the .05 level (2-tailed), and correlations of .30 and larger are significant at the .01 level (2-tailed).

Table 3. In Table B1 in the appendix descriptive statistics and the minimum and maximum score can be found for the unstandardized data for all groups and for the total sample.

## Group comparison

The MANOVA revealed a significant overall effect of Group (Pillai's Trace = .96, $F$ (26, 13) = 4.43, $p < .001$, $\eta_p^2 = .48$). Pillai's Trace is reported because Box's test of equality of covariance matrices was significant ($M = 346.64$, $F$ (282, 14371.80) = 1.42 $p < .001$). For all measures the effect of Group was significant ($p < .05$). Pairwise comparisons with the Least Significant Difference method were performed after the MANOVA; effect sizes and significances can be found in Table 4. The HRDys group performed significantly worse than the LRnonDys control group on all measures, and effect sizes were especially large for the reading fluency and spelling measures. The HRDys group also performed significantly worse than the HRnonDys group on most measures, except for the digit span forward task and Dutch vocabulary, although the difference was marginally significant ($p = .07$) for the latter test. Although the HRnonDys group always scored in between the HRDys and LRnonDys group, their performance was in most cases not significantly different from the LRnonDys group. Exceptions here are the Dutch pseudoword reading fluency, spoonerisms RT and the digit-span Forward tasks where the HRnonDys group performed significantly worse than the LRnonDys group. For English spelling there was a marginally significant difference between the two non-dyslexic groups ($p = .09$).

## Comparison between English and Dutch literacy skills

As can be seen from the correlations in Table 3, English and Dutch WRF and spelling skills were overall strongly correlated, while there was only a small correlation between Dutch

**Table 4 Effect sizes and confidence intervals of group differences.**

| Measure | LRnonDys–HRDys | | | HRnonDys–HRDys | | | LRnonDys–HRnonDys | | |
|---|---|---|---|---|---|---|---|---|---|
| | Hedges' $g$ | 95% Confidence interval | | Hedges' $g$ | 95% Confidence interval | | Hedges' $g$ | 95% Confidence interval | |
| | | Lower bound | Upper bound | | Lower bound | Upper bound | | Lower bound | Upper bound |
| Dutch Words RF | 2.25*** | 1.55 | 2.94 | 1.86*** | 1.20 | 2.52 | .25 | −.30 | .80 |
| Pseudoword RF | 2.65*** | 1.91 | 3.40 | 1.79*** | 1.13 | 2.44 | .75*** | .19 | 1.31 |
| Loanword RF | 1.59*** | .97 | 2.22 | 1.25*** | .64 | 1.86 | .48 | −.07 | 1.03 |
| English Word RF | 1.60*** | .98 | 2.23 | 1.20*** | .59 | 1.80 | .52 | −.03 | 1.07 |
| Dutch Spelling | 1.77*** | 1.12 | 2.41 | 1.37*** | .76 | 1.99 | .50 | −.05 | 1.06 |
| English Spelling | 2.42*** | 1.71 | 3.14 | 1.98*** | 1.30 | 2.65 | .52* | −.04 | 1.07 |
| Dutch Vocabulary | .70*** | .14 | 1.26 | .61* | .04 | 1.18 | .25 | −.29 | .80 |
| English Vocabulary | .71*** | .15 | 1.27 | .57** | .01 | 1.14 | .20 | −.35 | .74 |
| Spoonerisms - Accuracy | 1.70*** | 1.06 | 2.33 | 1.13*** | .53 | 1.72 | .66 | .10 | 1.21 |
| Spoonerisms - RT (Log) | −2.79*** | −3.55 | −2.02 | −1.45*** | −2.07 | −.82 | −1.18*** | −1.76 | −.59 |
| RAN (Items/min) | .94*** | .37 | 1.51 | .70** | .13 | 1.28 | .20 | −.34 | .75 |
| Digit Span - Forward | 1.20*** | .61 | 1.79 | .54 | −.03 | 1.10 | .74*** | .18 | 1.31 |
| Digit Span - Backward | 1.02*** | .44 | 1.59 | .57** | .01 | 1.14 | .34 | −.21 | .89 |

**Notes.**

The indicated statistical significance is based on pairwise comparisons using the Least Square Difference method in the MANOVA.

*$p < .01$.

**$p < .05$.

***$p < .1$.

LRnonDys, Low Risk without Dyslexia; HRDys, High Risk with Dyslexia; HRnonDys, High Risk without Dyslexia; RF, Reading Fluency; RT, Reaction Time; RAN, Rapid Automatized Naming; MANOVA, Multivariate Analysis of Variance.

and English vocabulary. To compare Dutch and English WRF, spelling and vocabulary between groups, we performed a 2 × 3 repeated measures ANOVAs with Language (English vs. Dutch) as within-subject factor, and Group (LRnonDys vs. HRDys vs. HRnonDys) as between subject factor. For WRF there was a significant main effect of Group ($F$ (2, 74) = 35.09, $p < .001$, $\eta_p^2 = .49$), and a significant interaction between Language and Group ($F$ (2, 74) = 5.15, $p = .01$, $\eta_p^2 = .12$). There was no main effect of Language ($F$ (1, 74) = .35, $p = .56$, $\eta_p^2 = .01$). A main effect of Language could not be expected in this analysis, since the data were normalized per subtest and thus per language. Pairwise comparisons using the LSD method revealed that the HRDys group overall performed poorer than the LRNonDys and HRNonDys groups ($p < .05$), but that the nonDys groups did not differ from each other ($p > .05$). Further analyses of the interaction showed that the difference between English and Dutch WRF (calculated by subtracting the Dutch WRF score from the English WRF score) was larger for the HRDys group ($M = .44$, $SE = .14$) compared to the LRnonDys group ($M = −.03$, $SE = .14$, $p = .04$) and HRnonDys group ($M = −.26$, $SE = .18$, $p = .002$). Thus, the HRDys group performed relatively well on English compared to Dutch WRF. In Fig. 1, that shows the means and standard errors per group per language, this interaction can also be observed. The difference between the LRnonDys and HRnonDys groups was not significant ($p = .30$).

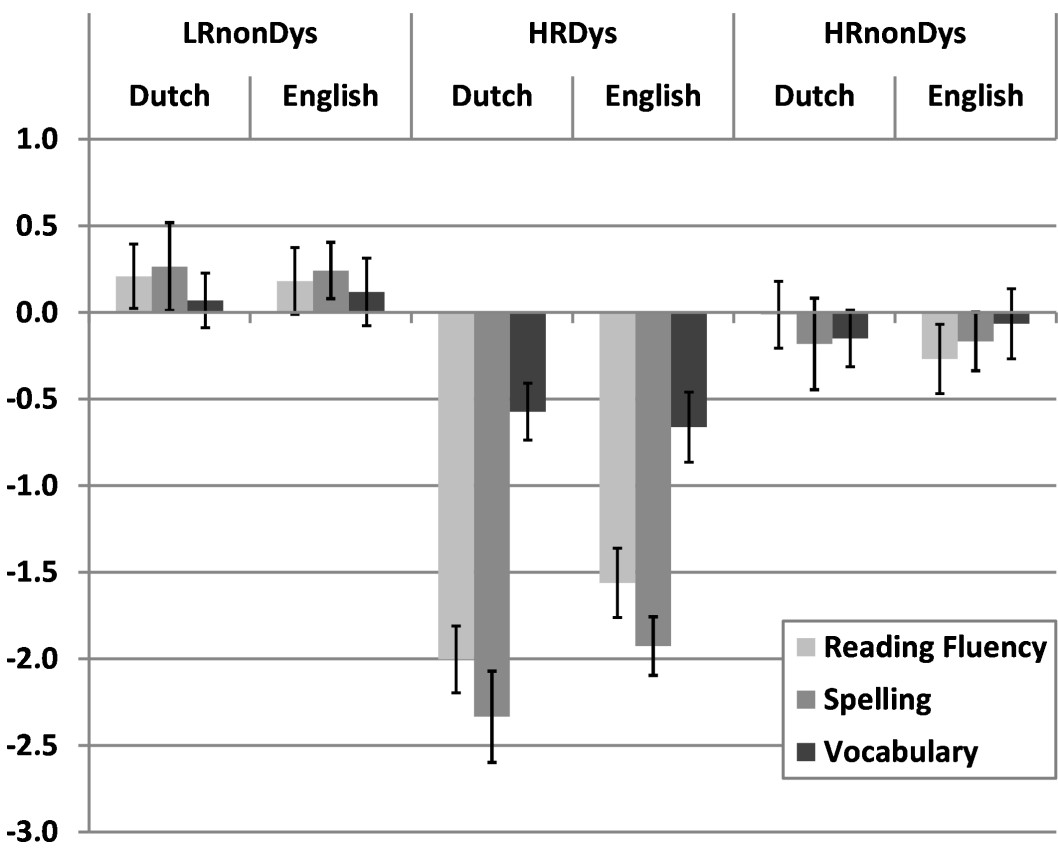

**Figure 1 Mean standardized scores on Dutch and English word reading fluency, spelling and vocabulary for the three groups.** Error bars indicate standard errors. (LRnonDys, low familial risk without dyslexia; HRnonDys, high familial risk without dyslexia; HRDys, high familial risk with dyslexia).

There also was a main effect of Group on spelling ($F (2, 74) = 41.36, p > .001, \eta^2_p = .53$) and vocabulary ($F (2, 74) = 6.34, p = .003, \eta^2_p = .15$). Pairwise comparisons for both spelling and vocabulary revealed that the HRDys group performed poorer than the LRNonDys and HRNonDys groups ($p < .05$), but that both nonDys groups did not differ from each other ($p > .05$). In contrast to WRF, there was no interaction between Language and Group for spelling ($F (2, 74) = 1.58, p = .21, \eta^2_p = .04$) or vocabulary ($F (2, 74) = .17, p = .84, \eta^2_p = .005$). There was also no main effect of Language on spelling ($F (1, 74) = .1.48, p = .23, \eta^2_p = .02$) or vocabulary ($F (1, 74) = .01, p = .91, \eta^2_p > .000$), but again this was expected because the data were standardized per subtest.

## DISCUSSION

### Reading skills of adolescents (with a familial risk of) dyslexia

The first aim of this study was to investigate how adolescents with a familial risk of dyslexia perform at reading and spelling tasks in Dutch and ESL in comparison to a low familial risk control group without dyslexia. Moreover, we compared these groups on several literacy related measures: English and Dutch vocabulary, RAN, PA, and VST/WM. Large reading

and spelling deficits and mild vocabulary deficits were found in the HRDys group, both in Dutch and in English. The HRDys group scored significantly lower than the LRnonDys Group on all other reading-related tasks as well. The HRDys group also scored significantly worse than the HRnonDys group on all measures with the exception of Dutch vocabulary, where there was a marginally significant difference. These findings are in agreement with previous studies that also found that dyslexia is persistent into adolescence (*Ferrer et al., 2015*; *Shaywitz et al., 1999*).

We observed a three-step pattern in the data as the scores of the HRnonDys group were in between the scores of the LRnonDys group and the scores of the HRDys group, but the difference between the two nonDys groups was only significant for Dutch pseudoword reading, the Spoonerisms RTs measuring Dutch PA and the digit span-forward task measuring VSTM. The results are in line with studies of adolescents with a familial risk where the HRnonDys group generally performs poorer than the LRnonDys group, but only some of the differences were significant (*Dandache, Wouters & Ghesquière, 2014*; *Eklund et al., 2014*; *Snowling, Muter & Carroll, 2007*). These results support a continuous view of (the familial risk of) dyslexia and a multifactorial origin of dyslexia as described by among others *Pennington et al. (2012)* and *Van Bergen, Van der Leij & De Jong (2014)*. It is likely that adolescents in the HRnonDys group have been exposed to some of the same family risk factors that affected the HRDys group. However, based on this study we cannot determine whether these family influences are genetic, environmental, both, or an interaction between genetics and the environment. This view of dyslexia is further supported by family-risk studies where the reading skills of parents of high FR children who did develop dyslexia have been found to be even lower than the reading skills of parents of high FR children who did not develop dyslexia (*Torppa et al., 2011*; *Van Bergen et al., 2011*; *Van Bergen et al., 2012*).

The effect sizes of the group differences were highly variable. For the dyslexic group compared to the nondyslexic groups, effect sizes were especially large for reading and spelling measures, as would be expected based on the definition of dyslexia. For the reading related measures, effect sizes were especially large for Dutch PA, and in particular for the reaction times. Also, the largest difference between the HRnonDys and LRnonDys group was on this PA measure. These findings are in line with a phonological core deficit in dyslexia (*Bekebrede, Van der Leij & Share, 2009*; *Van der Leij & Morfidi, 2006*). Although it has been shown that phonological problems are neither necessary nor sufficient to explain dyslexia (*Pennington et al., 2012*), many children with dyslexia do have a severe phonological deficit. Our study suggests that also non-dyslexic adolescents with a high FR have phonological problems, measured with Dutch PA, although milder than students with dyslexia. While the HRnonDys group generally scored lower than the LRnonDys group, their RAN, Verbal Working Memory (VWM) and both English and Dutch vocabulary scores were very similar, as indicated by the small effect sizes. For the HRnonDys group RAN, VWM and/or vocabulary knowledge may be protective factors or compensations for their phonological deficit; further research is needed to investigate whether the good RAN, VWM and vocabulary skills are a cause or consequence of their reading level, or whether they are merely correlated. These results are in line with *Moll, Loff & Snowling (2013)* who found that PA and VSTM were associated with family risk status as well as with

dyslexia status. The authors argued that phonological skills can be considered as a cognitive endophenotype of dyslexia. In addition, RAN, word recall and morphological awareness were also in their study only associated with dyslexia status, and were seen as additional risk factors for dyslexia. For both HR groups effect sizes were largest for pseudoword reading in comparison to the other reading and spelling tasks, probably because pseudoword reading relies mainly on letter-to-phoneme decoding, in addition to the use of sub-lexical orthographic units, but word-based orthographic strategies cannot be used to compensate for their phonological deficit.

## English versus Dutch

Our second aim was to make a direct comparison of reading, spelling and vocabulary skills in Dutch and English. Overall Dutch and English reading, spelling and vocabulary skills were significantly correlated; there were strong positive relationships between L1 and L2 reading and spelling, and small positive relationship between L1 and L2 vocabulary. Thus, adolescents with dyslexia had not only a literacy impairment in Dutch, but generally also in ESL. These findings are in agreement with the LCDH (*Ganschow et al., 1991*) which states that L1 reading skills are a strong predictor for L2 reading outcomes. Furthermore, previous studies have also observed reading problems in L2 among readers with a reading deficit in L1 (e.g., *Helland & Kaasa, 2005*; *Helland & Morken, 2016*; *Łockiewicz & Jaskulska, 2016*).

To our knowledge, our study was the first to make a direct comparison of Dutch and ESL in a group of adolescents with a familial risk of dyslexia. This did not only allow us to study whether deficits of the dyslexic group were similar across languages, but also whether the HRnonDys group, which by definition has no reading deficit in L1, has literacy problems in L2. We found that the HRDys group performed, relative to the LRnonDys control group, better on WRF in English compared to Dutch, however, this interaction effect was small; for spelling and vocabulary this effect was not found. The HRnonDys group did not show any significant differences between Dutch and English reading, spelling and vocabulary, as compared to this LRnonDys pattern. The relatively greater reading fluency in English than in Dutch in the HRDys group may seem surprising as the lower orthographic transparency of English makes reading in English more difficult, at least for beginning readers (*Seymour, Aro & Erskine, 2003*). Moreover, the adolescents in our study were unbalanced bilinguals who acquired English later than Dutch, and who were generally more proficient in Dutch, therefore they may be less able to compensate using semantics-based reading strategies.

However, other factors may be involved as well. Because our participants were not beginning readers in Dutch, they had presumably already developed a whole-word based reading strategy in Dutch, and perhaps they could already apply this strategy when they learned to read in English. Because of the lower orthographic transparency, whole-word reading is more successful than decoding in English. Perhaps the reading deficit of dyslexic readers in ESL was therefore smaller than expected based on their Dutch reading. Furthermore, not all readers with dyslexia have an orthographic processing deficit (*Van der Leij & Morfidi, 2006*). In fact, an earlier study by *Siegel, Share & Geva (1995)* found that English dyslexic readers had superior orthographic skills compared to normal readers, which may suggest that they rely more on the visual orthographic characteristics of a word

than on its phonological characteristics. This result has, however, not been replicated. It is unclear whether the dyslexics as a group show better orthographic skills, or that it is a characteristic for a substantial subgroup, as is predicted by the phonological-core variable-orthographic differences model (e.g., *Bekebrede, Van der Leij & Share, 2009*). To read successfully in English, a reader needs to identify larger orthographic units than in Dutch. Therefore having average to good orthographic skills is more beneficial for reading in English. Some readers with dyslexia may have been able to use orthographic strategies to compensate for their phonological deficit in ESL, like the readers with a preference for English in the study by *Miller-Guron & Lundberg (2000)*.

Another influencing factor responsible for the advantage of ESL for students with dyslexia relates to the massive exposure to English that (young) people have nowadays. As the use of English becomes more and more important for adolescents to participate in modern society, the motivation to learn English may be high. Perhaps reading in English is also less associated with the frustration individuals with reading difficulties experienced as beginning readers in Dutch, because, when they learned ESL, they had already developed some compensation strategies in Dutch. These hypotheses are still speculative and need further investigation, but they seem to be supported by a few recent studies. In a Norwegian study by *Brevik, Olsen & Hellekjær (2016)* it was found that 22 percent of the worst readers in Norwegian L1 had good reading skills in ESL. Most of these poor L1 but good L2 readers were boys in vocational education programs. A follow-up case study of five boys in vocational programs with poor L1 Norwegian reading skills but good ESL reading skills showed that they all played online games in English and also used English more than Norwegian for Facebook, music, TV and films (*Brevik, 2016*). Also *Sundqvist & Wikström (2015)* found that frequent online-gamers, who were again mostly boys, had the highest English grades, wrote better English essays in which they used more advanced vocabulary, and had a larger L2 vocabulary size. High exposure through multimedia and intrinsic motivation to learn English as a lingua franca in modern society are two possible explanations for the slight advantage we found in English for students with dyslexia compared to their mother tongue.

## Limitations and future directions

In the present study we have chosen to use a categorical approach towards dyslexia and familial risk. We have done so because categorical diagnoses are still used for clinical practice and for the comparability with other studies. However we do realize that both dyslexia and familial risk are continuous constructs as we have explained in the introduction. Because of the longitudinal design groups were not matched and we had no control over factors like the quantity and quality of the remediation, instruction in and exposure to English, and had only limited information available about these factors. Future cross-sectional studies could look further into these factors as they could be relevant in the context of our research question. Since this study was conducted at the end of a longitudinal design, it had a higher dropout rate than previous waves of the DDP program, leaving the groups smaller than we had hoped for. This affected the statistical power, and may be the reason why some of the small effects between the two non-dyslexic groups were not significant. However,

the present study indicates that the differences between the HRnonDys and LRnonDys groups are not large. For the comparison of Dutch and English vocabulary it would have been better if we used more similar tests. Because of time concerns, we used the vocabulary test from the WISC-III-NL (*Wechsler, 2005*) for Dutch and the more elaborate PPVT-IV (*Dunn & Dunn, 2007*) for English. This could at least partially explain why the correlations between the English and Dutch vocabulary tests were small. Moreover, a standardized spelling ESL test for Dutch learners was not available. So, we had to create a new test. However, because of strong correlations with the Dutch spelling test and the English WRF, we judged it sufficiently valid and reliable.

Including a larger sample would allow for the investigation of factors that contribute stronger to reading in Dutch versus reading in English, and whether this is the same for groups with a different risk and dyslexia status. Moreover, this would allow us to look at subgroups, or even at the individual level to investigate the influence of exposure and motivation in more detail. In the present study we did not look at orthographic processing, but it would be really interesting to include this in future studies, and to also consider higher level reading skills such as sentence and text comprehension, silent reading, and to measure reading related skills and pseudoword reading also in L2. Future studies could also use neuro-imaging techniques to investigate which brain networks are involved in reading in L1 and L2 in the different groups. Such research could also help to identify compensation mechanisms used by the HRnonDys group that allows them to read well despite the mild phonological processing difficulties that we found in the present study using the Dutch PA task. Knowing more about what contributes to reading success in ESL is not only important for theoretical purposes but could also be useful for future interventions.

## Conclusions and implications

We found that dyslexia is persistent in adolescents with a high familial risk of dyslexia. Although the reading deficit of the adolescents with dyslexia was slightly smaller in ESL than expected based on their Dutch WRF skills, adolescents with dyslexia generally also have large reading and spelling deficits in L2. Therefore it is important that they still receive help for this in secondary education and are granted access to measures like extra time in exams and the use of spellcheckers in L1 and L2. A phonological deficit was found in both groups with a familial risk, however it was less severe in the HRnonDys group who may also have used their better RAN- and VWM-abilities to compensate. Despite their subtle phonological deficit, and the low orthographic transparency of English, the HRnonDys group did not have reading difficulties in ESL. Thus, their familial risk seems to affect L1 and L2 in a similar way, and extra help in ESL is probably not needed for most of these students.

## ACKNOWLEDGEMENTS

We thank the master students who helped with the data collection. We are also thankful for the participation of the children and the commitment of their parents to this study.

### Funding
This work is part of the research programme 360-89-040, which is financed by the Netherlands Organisation for Scientific Research (NWO). There was no additional external funding received for this study. The funders had no role in study design, data collection and analysis, decision to publish, or preparation of the manuscript.

### Grant Disclosures
The following grant information was disclosed by the authors:
Netherlands Organisation for Scientific Research (NWO): 360-89-040.

### Competing Interests
The authors declare there are no competing interests.

### Author Contributions
- Ellie R.H. van Setten conceived and designed the experiments, performed the experiments, analyzed the data, contributed reagents/materials/analysis tools, wrote the paper, prepared figures and/or tables, reviewed drafts of the paper.
- Wim Tops conceived and designed the experiments, contributed reagents/materials/analysis tools, wrote the paper, reviewed drafts of the paper.
- Britt E. Hakvoort conceived and designed the experiments, performed the experiments, reviewed drafts of the paper.
- Aryan van der Leij conceived and designed the experiments, contributed reagents/materials/analysis tools, reviewed drafts of the paper.
- Natasha M. Maurits conceived and designed the experiments, reviewed drafts of the paper.
- Ben A.M. Maassen conceived and designed the experiments, analyzed the data, reviewed drafts of the paper.

### Human Ethics
The following information was supplied relating to ethical approvals (i.e., approving body and any reference numbers):
The study was approved by the Medical Ethical Committee of the University Medical Center Groningen. (METc-nr: 2014/076, ABR-nr: NL48140.042.14).

### Data Availability
The raw data has been provided as a Supplemental File.

### Supplemental Information
Supplemental information for this article can be found online at http://dx.doi.org/10.7717/peerj.3895#supplemental-information.

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
