# Peer review of "L1 and L2 reading skills in Dutch adolescents with a familial risk of dyslexia"

_PeerJ, doi:10.7717/peerj.3895_

## Round 0.1 · original submission · Major Revisions

We have now received the reports from two Reviewers. Both are quite positive, and they have offered helpful comments on your manuscript with regard, in particular, to experimental design. I agree with these comments and would expect changes along the lines suggested by the Reviewers if you choose to re-submit. Both agree that you have gone beyond the data with regard to the polygenetic basis of dyslexia, and alternative suggestions have been recommended.

Reviewer 1 ·

Basic reporting

The article is well written, covers most of the relevant literature (except see comment on Siegel’s work) sets out the questions clearly. See also comments on raw data below.

Experimental design

I understand that this journal places a premium on technical adequacy rather than conceptual significance, but at the core of this article lies a pointless analysis, the comparison of the HRDys group with the other two groups. Whatever the technical adequacy of the analysis, it is pointless because the groups were selected to be distinct on reading in the first place, so finding a “significant” difference is all but a foregone conclusion. Further, with the well-attested high correlation between reading and spelling, the latter could equally be expected to show a significant difference. It’s less certain but reasonable guess that they will be poorer in commonly related cognitive processes like PA. If the article is to be published, the authors should drop that aspect of their analysis.

More worthy of consideration are the questions of how children classified as dyslexic in Dutch fare in English, and how the non-dyslexic family-risk group fare in relation to their non-FR counterparts. The first of these questions was addressed by the subtraction statistic (Dutch WRF – English WRF), with the finding of a greater difference for the HRDys group than the other two groups. The only threat to this counterintuitive finding that I can see would be ceiling effects for the LRnonDys and HRnonDys groups (such as many of these participants reading all the words in the allotted time). Since the authors do not present non-standardized scores, they should address, and hopefully eliminate, this possibility. Supplying raw data would also help readers appreciate is there are any limitations to this finding along the lines that I have speculated.

In their discussion of this particular result they might also refer to a finding by Linda Siegel to the effect that RD children can be superior in the “visual route,” to use dual-route terminology.

The comparison of the two nonDys groups suffers from low power, I believe, leaving uncertainty about whether there is a population difference between them for the variables that were non-significant. I think the authors would be entitled to use one-tailed tests of significance because they are making directional predictions, which may clarify matters, and in any case their findings are like many others’ and support the continuous nature of dyslexia risk. It’s just a pity that they did not address the question of power and, if possible, design their study accordingly (or use a Bayesian approach based on prior probabilities of difference). I also add that it is going well beyond the data to assume that the results support the polygenic account of dyslexia, that “the HRnonDys group are likely to have inherited some but not all genetic risk factors that contribute to the development of a reading impairment” (lines 431-432). We have no way of knowing this, and it is equally likely that the difference is down to environmental compensation in the nonDYS group, or a range of other possibilities.

Validity of the findings

No comment on top of those in previous section.

·

Basic reporting

Basic reporting, overall, is of good quality. The following, however, are recommended improvements:

Ambiguities
(i) Lines 150 & 176: As these pertain to reading, "sound-to-letter conversions" (or "phoneme-to grapheme") are not relevant. Letter-to-sound (or grapheme-to-phoneme) conversions are relevant.
(ii) 254-257: If the children read aloud, this should be stated. Although labelled as tests of "fluency," they measured the difficulty level of reading accuracy as well as speed of reading. They are tests of "accuracy/fluency."
(iii) 416, 426, 439-441: The text should make it explicit that the Spoonerism measures of PA are for LI (Dutch only).
(iv) 551, 560: It would be helpful to the reader if the "phonological problem" (or "deficit") were specified as either PA or pseudoword reading, and that neither were obtained for L2 (English).

Missing Relevant Background Information
An obtained result was that high-risk dyslexics, relative to their L1 (Dutch) word reading levels, had a positive advantage in L2 (English) word reading levels (Lines 480-482). In view of this, inclusion of the ages of the participants are recommended for the two previous studies (Miller-Guron & Lundberg,175-178, and van der Leij & Morfidi, 178-181) that gave results of some similarity. In addition, it is recommended that there be information about the method in the van der Leij & Morfidi study by which their subgroup "was shown to have superior orthographic competence" (178-181). If Miller-Guron & Lundberg obtained research evidence about the influence of irregular conversion rules in their study, then it is also recommended that information about their research method be included.

Minor Errors in the Text
See lines 296, 298, 459, and 477

Experimental design

The research design fills a knowledge gap and the method has been well executed. The statistical analyses are of a high standard.

Lines 521-532 should be included in the Participants section (200- ). Although the instructional and remedial history of the participants was "uncontrollable" (526-527), it should be described in summary form for each of the three main groups. This is of some interest in view of lines 558-560 and 563-565 of the Conclusions.

Validity of the findings

(i) The statistical analyses have been appropriately interpreted to support the findings.
(ii) However, in 429-432, this "view of" the "polygenetic origin of dyslexia" may be consistent with the results but, without empirical genetic evidence in the study, so also would be a view that there was only familial influence with no significant genetic origin.
(iii) The expressed limitations and future directions are pertinent to the obtained results.

Additional comments

For testing the "orthographic strategy" speculations (lines 457-461, 492-497, 545-546) the authors should attempt an item analysis of their existing data for the participants' reading of L2 (English) words with only regular grapheme-phoneme correspondences, in comparison with those having one or more non-regular correspondences. If there were sufficient non-regular words in the L1 (Dutch) word reading, the same analysis could be applied to that.

---

## Round 0.2 · Minor Revisions

Two Reviewers have now considered and commented favorably on your revision. Reviewer 2 has asked for some minor revisions as per your Rebuttal letter. I agree with him as it would add more clarity to the discussion of your results.

Reviewer 1 ·

Basic reporting

No further comments

Experimental design

No further comments

Validity of the findings

No further comments

Additional comments

The authors have attended well to the worries that I expressed in my original review. I now believe that the article meets the requirements of the journal, and support publication. Congratulations on a nice piece of research.

·

Basic reporting

The appropriate improvements have been made in this Revision of July 2017.

Experimental design

The changes in this area are also as in 1.

Validity of the findings

In their Rebuttal Letter the authors state that their study "cannot determine whether this familial influence is genetic or environmental." Unfortunately this revision, p. 20, lines 18-22, fails to clearly say this. It is obscured by such terms as "multicausal origin" (their text reads "multiclausal"), "may have," and "support." This reviewer approves the authors clear statement of their Rebuttal Letter, which should replace these obscurities. Anything else would not meet the required standard.

---

## Round 0.3 · accepted · Accept

I enjoyed reading your final revisions and I believe your research will make a useful and important addition to our knowledge of the area.

·

Basic reporting

No comment

Experimental design

No comment

Validity of the findings

In this revision, this area now meets the standard required.